# Optic nerve sheath diameter in patients with hepatic encephalopathy

**Nese Colak** [1]*, **Ozlem Bayrak Basakci**[2], **Basak Bayram**[1], **Ersin Aksay**[1], **Muhammet Kursat Simsek**[3], **Nuri Karabay**[4]

1 Department of Emergency Medicine, Dokuz Eylül University School of Medicine, Izmir, Turkey,
2 Department of Emergency Medicine, Okan University School of Medicine, Istanbul, Turkey, 3 Department of Radiology, Manisa Provincial Health Directorate Merkezefendi Public Hospital, Manisa, Turkey,
4 Department of Radiology, Dokuz Eylül University School of Medicine, Izmir, Turkey

* nese.oray@deu.edu.tr

## Abstract

### Background

This study aims to reveal whether the optic nerve sheath diameter (ONSD) increases in hepatic encephalopathy (HE) patients, and to determine ONSD is associated with the poor prognosis of patients with HE.

### Methods and material

In this retrospective case-control study, HE patients who underwent cranial computerized tomography (CT) were included in the case group; and the patients who underwent CT for other reasons for the same age and gender and were normally interpreted were included in the control group. ONSD measurements in the case and control groups and clinical grades of HE with in-hospital mortality and ONSD measurements were compared in the case group.

### Results

This study was done with 74 acute HE patients and 74 control patients. The mean age was 62.9 ± 11.0 years and 67.6% of patients were male in both groups. The ONSD in the case group was higher than the control group (5.27-mm ± 0.82 vs 4.73 mm ± 0.57, $p$ <0.001). In the case group, the ONSD was 5.30 mm ± 0.87 in survivors, and 5.21 ± 0.65 in non-survivors (**$P$** = 0.670). There was no significant difference between the West Haven HE grade (**$P$** = 0.348) and Child-Pugh Score (**$P$** = 0.505) with ONSD measurements.

### Conclusion

We have shown that ONSD increases in HE patients compared to the control group. ONSD was not related to the Child-Pugh Score, HE grade, and in-hospital mortality.

**Data Availability Statement:** All relevant data are within the paper and its Supporting Information files.

**Funding:** The authors received no specific funding for this work.

**Competing interests:** The authors have declared
that no competing interests exist.

## Introduction

Hepatic encephalopathy (HE) is a neuropsychiatric syndrome in which a decreased level of consciousness, attention, and personality changes are observed due to the liver's inability to process toxic metabolites. It has been shown that 80% of chronic liver failure patients ultimately will develop mild HE, and 30–40% develop severe HE [1].

In cases that involve an increase in intracranial pressure (ICP), the subarachnoid area and retrobulbar segments are affected and the optic nerve sheath diameter (ONSD) increases [2]. It has been shown that there is a positive correlation between ONSD and an increase in ICP, however, most of these studies have examined structural etiologies of increased ICP such as intracranial edema, intracranial hemorrhage, and ischemic cerebrovascular accident [2–5]. On the other hand, an increase in ICP is also seen in a metabolic coma, such as hypertensive encephalopathy, diabetic ketoacidosis, and HE [6–10]. One study has also reported that increased ONSD is correlated with the degree of HE in acute liver failure in children in a study, but this association has not yet been demonstrated in adults [11].

There are several important hypotheses regarding the increase in intracranial pressure in HE. The most accepted hypothesis is that ammonia detoxifies glutamine in astrocytes and glutamine causes swelling and brain edema development in astrocytes by osmotic effect. Cytotoxicity in astrocytes causes disruption of the blood-brain barrier. The other hypothesis is that cerebral edema develops secondary to cerebral vasodilation in HE patients [12, 13].

Head CT is frequently requested in HE patients due to a decrease in the level of consciousness, therefore ONSD can also measure readily in those patients. Our primary aim in this study is to evaluate the ONSD in patients with HE compared to the control group, and our secondary aim is to examine whether there is a relationship between ONSD and in-hospital mortality and severity of HE grades in the case group.

## Materials and methods

### Study setting and design

This single-center, retrospective case-control study was conducted in a tertiary referral hospital emergency department (ED), with an approximately 200,000 annual admission rate.

### Selection of participants

Patients who were admitted to the ED between January 2014 and July 2018, diagnosed with HE, and undergoing brain CT were included in this study as the 'case group'. For the case group, patients with serum ammonia levels above the cut-off value (> 90 ug/dL) and who were consulted with gastroenterology considering hepatic encephalopathy according to their clinical findings were identified from the hospital information management system. Of these patients, those who had a head CT scan were included in the study. To determine the control group patients, the patients who had head CT in the ED during the study period were evaluated in chronological order. Patients of similar age (age of study group ± 2) and gender, with normal head CT performed for an indication other than altered mental status (most of those patients were presented to the ED with syncope, headache, and mild head trauma) and without known hepatic failure were included in this study as the 'control group'. The following patients were excluded from the study groups: (1) patients with intracranial structural lesions that increase ICP, such as intracranial mass, intracranial hemorrhage; (2) patients with moderate to severe head trauma at presentation or a history of previous intracranial surgery; (3) patients who are not clinically considered to have HE despite high ammonia levels (patients with normal consciousness or patients receiving

valproic acid therapy). If the patient had more than one admission from the ED for a diagnosis of HE, only the first admission was included in the study.

## Data collection

The study data collection was conducted from 15th August 2018 to 30th December 2018. Demographic data, the etiology of cirrhosis, blood ammonia level, other laboratory values, in-hospital mortality, West Haven HE grade, and Child-Pugh classification were collected from the records in the hospital information management system. ONSDs were measured from CTs on the Picture Archiving Communication System of the hospital by one radiologist.

## Methods of measurement

CT scans were obtained from the 160-slice CT scanner (Toshiba® CT Aquilion Prime,). A neuroradiologist that blinded to the patient's clinical characteristics and outcomes was measured ONSDs. Measurements were made from both eyes, in the axial sections of CTs, 3 mm behind the optic disc (Fig 1). The mean of the right and left eye ONSD was taken as the average ONSD value for final analysis.

The severity of HE was graded with the West Haven Criteria and Child-Pugh classification. West Haven criteria categorize HE from grade 1 to grade 4. Grade 1: Mild lack of awareness, some subtle personality changes, and shortened attention span, Grade 2: Lethargy, disoriented, inappropriate behavior, obvious asterisks and slurred speech, Grade 3: Sleepy but somnolent but arousable, gross disorientation, inappropriate behavior, muscle rigidity, clonus and hyperreflexia, Grade 4: Coma and cerebral posture.

In the Child-Pugh classification, the degree of ascites (None = 1 point, slight = 2 points, moderate = 3 points), bilirubin level ($< 2$ mg/ml = 1 point, 2 to 3 mg/ml = 2 points, $> 3$ mg/ml = 3 points), albümin level ($> 3.5$mg/ml = 1 point, 2.8 to 3.5mg/ml = 2 points, $< 2.8$mg/ml = 3 points) and the prothrombin time ($< 4$ sec = 1 point, 4 to 6 sec = 2 points, $> 6$ sec = 3 points), and the degree of encephalopathy (None = 1 point, Grade 1 and 2 = 2 points, Grade 3 and 4 = 3 points) were evaluated. A child-Pugh score of 5 to 6 is was considered as class A, 7 to 9 is class B, and 10 to 15 is class C.

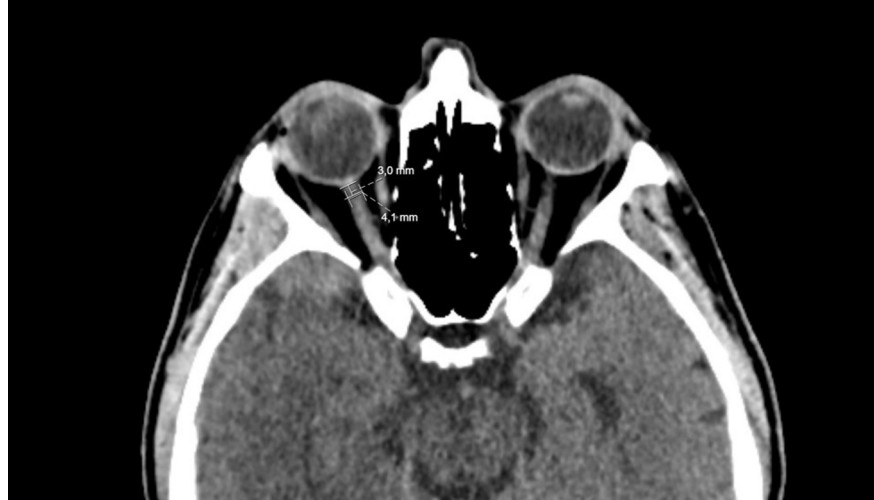

**Fig 1. Sample image showing that ONSD measurement (69 years old male patient with grade 3 HES).**

### Ethics statement

Approval for the study was obtained from the Ethics Committee of XXX University, School of Medicine (decision date 09.08.2018, no: 2018 / 21–06). The research is in line with the Helsinki Declaration of 1975.

### Statistical analysis

The data were analyzed in the SPSS 24.0 (Statistical Package for Social Sciences for Windows) program. The normality of the data was analyzed with the Kolmogorov-Smirnov test and the variance homogeneity was evaluated with the Levene test. Comparison of continuous variables between groups was analyzed with Student t-Test or Mann Whitney U test according to the distribution of the variables. The proportions of categorical variables in independent groups were evaluated using the Chi-Square test. Pearson correlation coefficients were calculated to examine the correlation between data groups. Variables were analyzed at a 95% confidence interval and $P$ <0.05 was considered as significant.

## Results

During the study, 384 patients with high ammonia levels and performed head CT scan in the ED were detected. A hundred and five patients without a clinical diagnosis of hepatic encephalopathy, 65 patients with other pathologies that may lead to increased ICP on CT, 10 patients with cranial trauma at admission, and subsequent admissions of 21 patients who were admitted more than once to the ED were excluded from the study. Finally, we analyzed 74 HE patients and 74 patients in the control group. The mean age of patients was 62.9 ± 11.0 years, and 67.6% (n = 50) of the patients were male in both groups.

### Primary outcome

The mean ONSD of the case group was higher than the control group (5.27 ± 0.82 mm vs 4.73 ± 0.57 mm, $P$ <0.001). When the right eye and left eye ONSD measurements were evaluated separately, the average ONSD in the case group was higher than in the control group.

There was no difference between in the right and left ONSD measurements in the case group (5.29 ± 0.79 mm and 5.26 ± 0.93 mm, $P$ = 0.602) and the control group (4.76 ± 0.56 mm and 4.70 ± 0.64 mm, $P$ = 0231) (Table 1). There was a good correlation between right and left ONSDs in both groups (r = 0.804 for the case group and r = 0.811 for the control group).

### Patient characteristics of the case group

The most common etiology of cirrhosis was Alcohol (27%), chronic hepatic B infection (20%), and chronic hepatic C infection (15%). The West Haven HE grades, Child-Pugh scores, and mortality rates with the ONSD measurements are shown in Table 2. In the case group, 18

**Table 1. ONSD results of the case and control groups.**

| ONSD measurements | Case group mean±SD (min-max) | Control group mean±SD (min-max) | P value |
|---|---|---|---|
| **Average ONSD**[*] | 5.27±0.82 (3.8–8.3) | 4.73±0.57 (3.5–6.2) | <**0.001** |
| **Right ONSD** | 5.29±0.79 (3.8–8.3) | 4.76±0.56 (3.2–6.1) | <**0.001** |
| **Left ONSD** | 5.26±0.93 (3.6–9.2) | 4.70±0.64 (3.0–6.3) | <**0.001** |

[*] Average ONSD = The mean of right and left ONSD measurements

ONSD: optic nerve sheath diameter, SD: Standart deviation

**Table 2. Patient characteristics of the case group.**

|  | All cases (n = 74) | Survive (n = 56) | Non survive (n = 18) | *P* value |
|---|---|---|---|---|
| **ONSD (mm) mean±SD** | 5.27±0.82 | 5.30±0.87 | 5.21±0.65 | 0.708 |
| **Child Pugh Score; Class A+B n (%)** | 33 (45) | 29 (88) | 4 (12) | 0.028 |
| **Child Pugh Score; Class C n (%)** | 41 (55) | 27 (66) | 14 (34) |  |
| **HE; Grade 1+2 n (%)** | 50 (68) | 41 (82) | 9 (18) | 0.067 |
| **HE; Grade 3+4 n (%)** | 24 (32) | 15 (63) | 9 (38) |  |
| **Ammonia (ug/dL) mean±SD** | 247±103 | 260±102 | 208±98 | 0.062 |
| **Glucose (mg/dL) mean±SD** | 146±61 | 147±64 | 143±50 | 0.790 |
| **Creatinine (mg/dL) mean±SD** | 1.5±1.0 | 1.4±1.0 | 1.8±1.0 | 0.69 |
| **Sodium (mmol/L) mean±SD** | 134±5 | 135±5 | 131±6 | 0.19 |
| **Potassium (mmol/L) mean±SD** | 4.4±0.8 | 4.3±0.6 | 4.8±1.1 | 0.008 |
| **Lactate (mmol/L) mean±SD** | 3.5±3.5 | 2.9±2.7 | 5.0±4.4 | 0.056 |
| **sGPT (IU/L) mean±SD** | 55±84 | 40±65 | 100±119 | 0.008 |
| **sGOT (IU/L) mean±SD** | 91±140 | 75±96 | 144±225 | 0.066 |
| **Albumin (g/dL) mean±SD** | 2.9±0.6 | 3.0±0.6 | 2.7±0.7 | 0.182 |
| **pH mean±SD** | 7.4±0.1 | 7.5±0.1 | 7.3±0.2 | 0.106 |
| **HCO$_3$ mean±SD** | 21.3±4.5 | 22.7±2.8 | 18.8±5.9 | 0.005 |
| **INR mean±SD** | 1.5±0.4 | 1.5±0.3 | 1.7±0.5 | 0.024 |

HE: hepatic encephalopathy, ONSD: optic nerve sheath diameter, SD: Standart deviation, sGOT: Serum glutamic oxaloacetic transaminase; sGPT: Serum glutamic pyruvic transaminase; INR: International normalized ratio.

(24%) patients were treated in the ED department and then discharged from the observation unit of ED, 56 patients were hospitalized. The in-hospital mortality rate was 24%.

There was no difference in age, gender, ONSD, etiology of cirrhosis, and HE grades between survivors and non-survivors. Potassium, Serum glutamic pyruvic transaminase (SGPT), International normalized ratio (INR) levels were higher and HCO$_3$ levels were lower in non-survivors than survivors. The clinical characteristics of the case group are shown in Table 3. Patients with Child-Pugh C grade had a higher mortality rate than those with Child-Pugh A and B grades.

## Secondary outcome

While the ONSD of the surviving patients was 5.30 ± 0.87 mm, the mean ONSD of the patients who died was 5.21 ± 0.65mm (*P* = 0.708) in the case group. According to the Child-Pugh and

**Table 3. ONSD measurements according to mortality, HE and Child-Pugh classifications.**

|  |  | n (%) | ONSD-*mm* mean±SD (min-max) | *P* value |
|---|---|---|---|---|
| Mortality | **Survive** | 56 (76) | 5.30±0.87 (3.8–8.3) | 0.708 |
|  | **Non-Survive** | 18 (24) | 5.21±0.65 (3.9–6.5) |  |
| HE | **Grade 1** | 13 (18) | 4.97±0.69 (3.8–6.3) | 0.348 |
|  | **Grade 2** | 37 (50) | 5.40±0.95 (3.9–8.3) |  |
|  | **Grade 3** | 18 (24) | 5.17±0.63 (4.0–6.1) |  |
|  | **Grade 4** | 6 (8) | 5.48±0.54 (5.0–6.5) |  |
| Child-Pugh Score | **Class A** | 2 (3) | 5.65±0.92 (5.0–6.3) | 0.505 |
|  | **Class B** | 31 (42) | 5.16±0.81(4.0–7.5) |  |
|  | **Class C** | 41 (55) | 5.35±0.83 (3.8–8.3) |  |

HE: hepatic encephalopathy, ONSD: optic nerve sheath diameter

West Haven HE grades, there was no difference between the ONSD of the patients (Table 3). There was also no significant correlation between serum ammonia level and ONSD measurements (correlation -0.036 and *P* = 0.759).

## Discussion

In this case-control study, we showed that the ONSD values in the head CT of patients with HE were increased compared to the control group. When the patients are grouped according to the severity of the HE, there seems to be a trend in lower ONSD measurements in grade 1 HE and the higher ONSD in grade 4 HE, however, this did not reach statistical significance and the trend was not consistent when including measurements across patients from grade I HE all the way through grade 4 HE. There was no significant relationship between the patients' Child-Pugh groups and ONSD measurements. There was also no difference between ONSD in survivors and non-survivors. In line with these data; it can be said that ONSD in HE patients is increased compared to the control group, but this increase was not related to the Child-Pugh Score, HE grade, and in-hospital mortality. Although ONSD is not a prognostic criterion in patients with hepatic encephalopathy, increased ONSD may help or support the diagnosis of hepatic encephalopathy.

In HE patients, it is thought that ICP increases due to cerebral vasodilation and edema. The pathophysiology of the development of cerebral edema is still unclear. The most widely accepted hypothesis is the "ammonia-glutamine" hypothesis. Increased ammonia in hepatic failure patients causes glutamine synthesis and accumulation in astrocytes. An elevated glutamine level increases intracellular osmolarity, astrocytes swell, and brain edema. Another hypothesis is the 'Trojan horse' hypothesis. According to this hypothesis, excess glutamine synthesized in astrocytes is transported to the mitochondria, where it is metabolized to ammonia and glutamate. Glutamine, the "Trojan horse", acts as the ammonia carrier to the mitochondria where its accumulation causes oxidative stress and ultimately astrocyte swelling [12–15].

The gold standard method for evaluating increased ICP is invasive ICP monitoring [16]. However, since invasive methods cannot easily be available in the ED setting, increased ICP can be estimated by indirect methods. In the case of increased ICP, the subarachnoid area and the retrobulbar segment are affected, and ONSD increases. Therefore, ONSD measurement is a non-invasive and reliable method used to estimate increased ICP. Although ultrasonography is more frequently preferred for ONSD measurement, there are many studies in which ONSD measurement is performed in CT and magnetic resonance imaging [4, 5, 17–19]. Head CT is often requested to exclude structural lesions in patients presented to the ED with HE. For this reason, ONSD can easily be evaluated in the currently performed head CT images and can give preliminary information about increased ICP. According to the meta-analysis of Koziarz et al, who examined the relationship between ONSD and increased ICP in the literature, increased ICP can be excluded with high sensitivity and a low negative likelihood ratio with ultrasonographic normal ONSD measurement (cut-off value was calculated as 5.0 mm in the study) [18]. In our study, the mean ONSD of the case group (5.27 ± 0.82) was higher than the control group (4.73 ± 0.57), which we can accept as the normal population.

Das et al. was reported that ultrasonographic ONSD measurements were useful in the monitorization of increased ICP in acute liver failure in children. In this study, the ONSD was correlated with HE grade, INR, and ammonia level. ONSD values were 4.2 mm in the control group, 4.4 mm in acute liver failure without HE, 5.2 mm in acute liver failure with HE, and 3.9 mm in those who recovered. As the grade of HE increased, ONSD levels were also increased, but this increase was not statistically significant. ONSD was higher in patients with hepatic

failure who developed HE compared to patients with liver failure who did not develop HE and a healthy population. They concluded that ONSD measurement may be useful in the diagnosis of HE in patients who are difficult to diagnose only with clinical evaluation [11]. ONDS may prove helpful to the child population, due to given various stages of speech and cognitive abilities making an assessment of encephalopathy difficult. Our study supports that ONDS may be helpful as a strong objective criterion for distinguishing hepatic encephalopathy in adults with impaired consciousness. This is a good initial study that explores the generalizability of a finding that has been more robustly demonstrated in the pediatric population.

Child-Pugh and West Haven classifications are the most frequently used scoring in determining the severity and prognosis of HE patients. In our study, because the number of patients in some groups in the West Haven and Child-Pugh classifications was low (for example, 3 patients in Child-Pugh grade A), we regrouped into 2-step categories. (HE grade 1 + 2 vs 3 + 4, Child-Pugh grade A + B vs C) However, there was no difference in ONSD measurements between the two-level groups. Although it did not reach statistical significance, patients with HE grade 1 had the lowest ONSD, while HE grade 4 patients had the highest ONSD. It is possible that due to the low sample size, this study was not powered enough to detect statistical significance. Future studies performed prospectively with a larger sample size may be able to detect a difference.

Diabetic ketoacidosis, like HE, is a cause of metabolic coma that may cause brain edema. Kendir et al. showed that an increase in ONSD in 31 children with diabetic ketoacidosis, in moderate-severe diabetic ketoacidosis, reached the highest diameter, and ONSD decreased after treatment. It has been reported that brain edema can be monitored by measuring ONSD during the treatment of these patients, and it can contribute to clinical treatments such as fluid therapy [7].

In our study, ONSD measurements were only made at the time of admission. For future studies, the change in ONSD measurements before and after HE treatment should be evaluated and its contribution to treatment guidance can be investigated. This should ideally be studied in a prospective study.

## Limitations

In our study, the single-center and study size was small to make firm conclusions about the association between ONSD and outcomes. In our study, ONSDs of survivors were higher than non-survivors. Mortality in study patients may be due to other causes (e.g. variceal bleeding, spontaneous bacterial peritonitis) rather than the severity of HE.

## Conclusion

We have shown that ONSD increases in HE patients compared to the control group. ONSD was not related to the Child-Pugh Score, HE grade, and in-hospital mortality.

## Supporting information

**S1 Data.**
(SAV)

## Author Contributions

**Data curation:** Nese Colak, Ozlem Bayrak Basakci, Basak Bayram, Ersin Aksay, Muhammet Kursat Simsek, Nuri Karabay.

**Formal analysis:** Nese Colak, Basak Bayram, Nuri Karabay.

**Methodology:** Nese Colak, Ozlem Bayrak Basakci, Basak Bayram, Ersin Aksay, Muhammet Kursat Simsek, Nuri Karabay.

**Project administration:** Nese Colak, Basak Bayram, Muhammet Kursat Simsek.

**Resources:** Nese Colak, Nuri Karabay.

**Software:** Muhammet Kursat Simsek.

**Supervision:** Nese Colak, Ozlem Bayrak Basakci.

**Visualization:** Ozlem Bayrak Basakci, Ersin Aksay.

**Writing – original draft:** Nese Colak, Ozlem Bayrak Basakci, Basak Bayram, Ersin Aksay.

**Writing – review & editing:** Nese Colak, Basak Bayram, Ersin Aksay.

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
