## [Decision Letter · Decision Letter 0]

24 Oct 2022

PONE-D-22-23776Does optic nerve sheath diameter increase in patients with hepatic encephalopathy?PLOS ONE

Dear Dr. Colak,

Thank you for submitting your manuscript to PLOS ONE. After careful consideration, we feel that it has merit but does not fully meet PLOS ONE’s publication criteria as it currently stands. Therefore, we invite you to submit a revised version of the manuscript that addresses the points raised during the review process.

We look forward to receiving your revised manuscript.

Kind regards,

Alok Arora, MD, FACP, FRCP

Academic Editor

PLOS ONE

Journal Requirements:

Reviewers' comments:

Reviewer's Responses to Questions

**Comments to the Author**

1. Is the manuscript technically sound, and do the data support the conclusions?

Reviewer #1: Partly

Reviewer #2: Yes

Reviewer #3: Yes

Reviewer #4: Yes

2. Has the statistical analysis been performed appropriately and rigorously? 

Reviewer #1: Yes

Reviewer #2: Yes

Reviewer #3: Yes

Reviewer #4: Yes

3. Have the authors made all data underlying the findings in their manuscript fully available?

Reviewer #1: Yes

Reviewer #2: Yes

Reviewer #3: Yes

Reviewer #4: Yes

4. Is the manuscript presented in an intelligible fashion and written in standard English?

Reviewer #1: Yes

Reviewer #2: Yes

Reviewer #3: Yes

Reviewer #4: Yes

5. Review Comments to the Author

Reviewer #1: - It is not appropriate to ask a question in the title of the article, change it

- In the Abstract Conclusion section, the authors said;"We have shown that ONSD increases in HE patients compared to the control group. and Our study demonstrates an association between increased ONSD in HE patients compared to the control group.'' Two sentences with the same meaning are written, please correct it..

- Specify West Haven HE grade and Child-Pugh classification in detail in the material method section. In addition, specify how many points you get from each criterion in these classifications in detail in the results section.

- Show on the figure the image of a real patient that you have measured the optic nerve diameter on CT.

-The authors stated that the in-hospital mortality rate was 24%. This rate seems too exaggerated to me. Is this rate for all patients diagnosed with Hepatic encephalopathy at the hospital or is it the rate of patients included in the study?

Reviewer #2: The manuscript is well written, in a very detailed and clear manner. The methods are explained clearly and the flow is good overall. The conclusion is consistent with the evidence presented. The study sample size is small but it suggested interesting correlation between hepatic encephalopathy and optic nerve sheath diameter. More studies need to be done in this regard, to determine its prognostic value in severe hepatic encephalopathy cases. I strongly recommend publication of this manuscript.

Reviewer #3: Your study is one of the first one done in adult population with liver cirrhosis and HE measuring ONSD with CT scan. I agree with your comment that your study may not have enough power to see any difference between different groups of Child Pugh Score or HE grade. It did not show any correlation in mortality outcome in liver cirrhosis patients but do you have any information on other parameters like length of stay which can be higher in more serious and complicated patients.

Reviewer #4: Decent article but the Etio pathogenesis should have been more clearly explained. Explanation of etio pathogenesis is very short , it needs to be further elaborated for a clear understanding of why ONSD increases in HE

6. PLOS authors have the option to publish the peer review history of their article (what does this mean?). If published, this will include your full peer review and any attached files.

Reviewer #1: No

Reviewer #2: **Yes: **Ramya Akella MD

Reviewer #3: No

Reviewer #4: **Yes: **Simhachalam Gurugubelli

---

## [Author Response · Author response to Decision Letter 0]

26 Oct 2022

Dear Editor;

We thank the Referees for their interest in our work and for their helpful comments on the manuscript and we have tried to do our best to respond to the raised points. We considered all comments and revised our manuscript according to your recommendations. All the changes marked as red on the marked copy. We explained how we changed the paper by answering the reviewer.

---

## [Editor Report · Decision Letter 1]

2 Nov 2022

Optic nerve sheath diameter in patients with hepatic encephalopathy

PONE-D-22-23776R1

Dear Dr. Colak,

We’re pleased to inform you that your manuscript has been judged scientifically suitable for publication and will be formally accepted for publication once it meets all outstanding technical requirements.

Kind regards,

Alok Arora, MD, FACP, FRCP

Academic Editor

PLOS ONE
---

## [Editor Report · Acceptance letter]

7 Nov 2022

PONE-D-22-23776R1 

Optic nerve sheath diameter in patients with hepatic encephalopathy 

Dear Dr. Colak:

I'm pleased to inform you that your manuscript has been deemed suitable for publication in PLOS ONE. Congratulations! Your manuscript is now with our production department. 

Kind regards, 

on behalf of

Dr. Alok Arora 

Academic Editor

PLOS ONE